# The Landscape of lncRNAs in Hepatocellular Carcinoma: A Translational Perspective

**DOI:** 10.3390/cancers13112651

**Published:** 2021-05-28

**Authors:** Juan Pablo Unfried, Paloma Sangro, Laura Prats-Mari, Bruno Sangro, Puri Fortes

**Affiliations:** 1Center for Applied Medical Research (CIMA), Department of Gene Therapy and Regulation of Gene Expression, Universidad de Navarra (UNAV), 31008 Pamplona, Spain; lpratsmari@alumni.unav.es (L.P.-M.); pfortes@unav.es (P.F.); 2Liver Unit, Clínica Universidad de Navarra (CUN), 31008 Pamplona, Spain; bsangro@unav.es; 3Navarra Institute for Health Research (IdiSNA), 31008 Pamplona, Spain; 4Liver and Digestive Diseases Networking Biomedical Research Centre (CIBERehd), 31008 Pamplona, Spain

**Keywords:** long non-coding RNAs, hepatocellular carcinoma, translational research, biomarkers, hallmarks of cancer

## Abstract

**Simple Summary:**

Hepatocellular carcinoma is the major form of liver cancer; it has a high incidence in the global population, and it is a leading cause of the world’s cancer burden. Despite great efforts to understand the disease at the molecular level and develop effective treatments for hepatocellular carcinoma patients, current therapies for advanced cases are only effective in a small percentage of patients. Therefore, it is paramount to discover new tumor targets that can be used to develop alternative therapeutic strategies. Promising novel targets could be long non-coding RNAs. These RNA molecules are frequently found to be involved in the development and progression of cancer. More importantly, they can be safely targeted with several strategies. However, our knowledge about long non-coding RNA biology and their clinical relevance remains underdeveloped. In this review, we summarize current efforts to validate the importance of long non-coding RNAs in hepatocellular carcinoma and to evaluate their potential to impact the clinical setting. To do that we highlight lncRNA implication in key events leading to cancer and we describe how affecting lncRNAs could broaden the repertoire of potentially useful targets to help meet the challenge of hepatocellular carcinoma treatment.

**Abstract:**

LncRNAs are emerging as relevant regulators of multiple cellular processes involved in cell physiology as well as in the development and progression of human diseases, most notably, cancer. Hepatocellular carcinoma (HCC) is a prominent cause of cancer-related death worldwide due to the high prevalence of causative factors, usual cirrhotic status of the tumor-harboring livers and the suboptimal benefit of locoregional and systemic therapies. Despite huge progress in the molecular characterization of HCC, no oncogenic loop addiction has been identified and most genetic alterations remain non-druggable, underscoring the importance of advancing research in novel approaches for HCC treatment. In this context, long non-coding RNAs (lncRNAs) appear as potentially useful targets as they often exhibit high tumor- and tissue-specific expression and many studies have reported an outstanding dysregulation of lncRNAs in HCC. However, there is a limited perspective of the potential role that deregulated lncRNAs may play in HCC progression and aggressiveness or the mechanisms and therapeutic implications behind such effects. In this review, we offer a clarifying landscape of current efforts to evaluate lncRNA potential as therapeutic targets in HCC using evidence from preclinical models as well as from recent studies on novel oncogenic pathways that show lncRNA-dependency.

## 1. Introduction

HCC is the most common primary liver cancer, with a global incidence of ~750,000 cases per year that nearly matches the mortality rate (700,000 deaths per year) [1,2]. Exposure to risk factors determines the incidence and mortality of HCC and it varies markedly between countries and regions. Advanced hepatic fibrosis or cirrhosis due to hepatitis B virus (HBV), hepatitis C virus (HCV), alcohol intake or the metabolic syndrome are the main well-established predisposing factors [3]. Aflatoxin B1 dietary intake is also a well-known environmental risk factor for HCC development in parts of Asia and Africa [4]. It is estimated that one-third of cirrhotic patients will develop liver cancer during their lifetime [5]. Strategies such as HBV vaccination, eradication of HCV with antiviral agents and community programs to improve lifestyle and eating habits have helped to reduce HCC predisposing factors. However, HCC remains the fourth leading cause of cancer-related deaths. Further, it has been projected that by 2030 the barrier of one million deaths per year will have been surpassed, consolidating HCC as the fastest-rising cause of cancer-related death worldwide [1,5,6,7,8]. Risk factors and factor-induced inflammation and fibrosis may induce DNA damage, epigenetic alterations or mutations leading to the activation of oncogenes or silencing of tumor suppressors, which might influence HCC development. These changes occur mainly in mature hepatocytes, but identification of up to 20% of HCCs with progenitor cell markers suggests that the cell of origin could also be a stem or progenitor cell in those cases [9,10].

Up to 50% of patients with HCC are diagnosed at early stages in developed countries, when radical treatments are feasible [11,12]. However, even though cirrhotic patients are monitored for early detection of HCC with ultrasound and/or alfa-fetoprotein (AFP) levels, many of them show advanced disease at diagnosis [11] or reach this stage after progression to local therapies. HCC is highly resistant to conventional cytotoxic chemotherapy. Sorafenib has been the only systemic treatment available for more than a decade [13]. Other tyrosine kinase inhibitors with an extended target spectrum but a similar antiangiogenic effect, including lenvatinib, regorafenib and cabozantinib, have joined sorafenib as alternative options for the first- or second-line therapy [3,11]. Immune checkpoint inhibitors that target cytotoxic T lymphocyte protein 4 (CTLA-4), programmed death receptor 1 (PD-1) or its ligand (PD-L1) have appeared as a new class of active agents against HCC. Nivolumab and pembrolizumab (anti-PD-1) are active in the second-line scenario [14,15]. Recently, the combination of atezolizumab (anti-PD-L1) with the vascular endothelial growth factor (VEGF)-inhibitor bevacizumab has proven superior to sorafenib in the first-line setting and is now the standard of care for patients naïve to systemic therapy [16]. During this time, several trials have shown no relevant antitumor activity using other agents with diverse molecular targets, including c-MET (tyrosine-protein kinase Met), mTOR (mechanistic target of rapamycin), EGFR (epidermal growth factor receptor) and others [17]. By sequential administration of available therapeutic agents, patients with a stably preserved liver function may survive for prolonged periods of time. However, most patients still do not respond to any of these therapies. The combination of agents targeting the immune response or cellular proliferation could further improve the outcome of HCC patients. However, this will require an exquisite understanding of the biology behind immunomodulation and HCC cell growth, including the identification of tumor-addictive targets within the coding and the non-coding genome.

Ordinarily, HCC studies that search for novel therapeutic strategies have been focused on protein targets. However, a global evaluation of the human genome indicates that only less than 2% accounts for coding sequences (exons that are translated into proteins) [18]. Hence, the non-protein-coding portion of the genome that is transcribed into non-coding RNAs (ncRNAs) has become increasingly valuable in recent years [19]. Out of all ncRNAs, long non-coding RNAs (lncRNAs) are the most numerous and the least studied. The relative novelty and general lack of knowledge of lncRNA functions so far, has prevented the development of classification systems that inform on lncRNA functional relevance or putative mechanisms of action. Therefore, major classifications describe lncRNA genes according to their location in the genome compared to their nearest protein-coding genes. In this context, lncRNAs are either intergenic (interspersed between coding regions) or intragenic (located within or very near to coding genes in either sense or antisense orientation). From a functional perspective, lncRNAs can be classified as *trans*-acting (which function similar to proteins, in cellular regions located away from the lncRNA transcription site) or *cis*-acting (which control the expression of neighboring genes co-transcriptionally, usually through transcriptional interference or chromatin modifications) [20]. Such localized function is exclusive to lncRNAs compared to proteins and it is an elegant and fast way of controlling a specific region of the genome without investing in DNA-binding domains. Other than that, lncRNAs, such as proteins, are folded into secondary and tertiary structures and may show domains capable of binding to DNA, other RNAs and/or proteins to form functional complexes and exert their functions. Most of the functions described to date are related to the regulation of coding genes at transcriptional and post-transcriptional levels [21]. Examples exist of lncRNAs that modulate DNA folding and enhancer activity, affect chromatin remodeling and epigenetic modifications and control transcription initiation or elongation. At the post-transcriptional level, lncRNAs have been described that can modulate splicing, translation and RNA stability by different means. In addition, some lncRNAs regulate protein function by influencing post-translational modifications or by acting as cofactors. Finally, although it seems contradictory, some lncRNAs encode small peptides with key functions in different cellular processes [22] (Figure 1). Interestingly, lncRNA function may be sequence dependent or independent. An excellent example of the latter is when the sole act of lncRNA transcription is required for function by transcriptional interference.

Transcription, processing, and regulation of most lncRNAs is very similar to that of their coding mRNA counterparts. Indeed, most lncRNAs are transcribed by polymerase II and are co-transcriptionally modified to undergo 5’capping, and oftentimes, polyadenylation, and splicing. However, splicing is less efficient and lncRNAs are more prone to undergo premature termination [23]. On average, lncRNAs have fewer exons, show less alternative splicing and are shorter than mRNAs. Other major differences are that, compared to mRNAs, lncRNAs are more nuclear, less abundant, more cell-type specific and, those deregulated in cancer, are far more tumor-type specific [24,25]. In fact, lncRNAs have been described as deregulated in various different cancers, including HCC, with implications as biomarkers for diagnosis, prediction of survival and risk of metastasis [26,27,28].

## 2. LncRNAs in the Hallmarks of HCC

Given the importance of lncRNAs, their ability to regulate gene expression at various levels and the differential profiles of lncRNA expression between human HCCs and non-tumoral tissue [29], lncRNAs are being actively studied for their potential as alternative therapeutic targets. In fact, the literature describes multiple lncRNAs deregulated in HCC with potential roles in therapeutically actionable pathways that strongly affect patient survival. Cell dedifferentiation, epigenetic alterations, chromatin instability and the activation of oncogenic pathways concomitant to hepatocarcinogenesis and tumor progression impact cell expression, leading to lncRNA deregulation. The altered lncRNA transcriptome may also result from increased inflammation and liver fibrosis caused by HCC risk factors. In the case of HCV and HBV patients, several lncRNAs can be deregulated in response to infection that may affect viral oncoproteins and increase hepatocarcinogenesis (reviewed in [30,31]). An interesting example is EGOT, a lncRNA induced after HCV infection that contributes to viral replication and HCC growth [32,33].

In the last decade, HCC–lncRNA research has grown exponentially, which has allowed the study of the increasing repertoire of mechanisms enabled by lncRNAs to impact the hallmarks of liver cancer. However, only a minority of studies have validated these findings in appropriate preclinical models and independent cohorts of patient samples to offer a clearer picture of their potential relevance for clinical translation. In this review, we have selected a collection of lncRNAs that have been published in high-impact journals, with thorough preclinical validation and/or association with relevant clinical parameters as a proxy for their clinical relevance. We are confident that this collection shows a clarifying landscape of the reality of lncRNA activity in HCC. Therefore, we have also grouped these lncRNAs in terms of their general mechanisms of action linked to the hallmarks of cancer they regulate.

In 2000, Hanahan and Weinberg proposed six hallmarks of cancer that all together constitute the key to unlock malignant transformation [34]. As it is well known, tumor formation is a multistage process where cells gain specific features and capacities that allow them to sustain proliferative signaling, evade growth suppressors, resist cell death, enable replicative immortality, induce angiogenesis, and activate invasion and metastasis. Ten years later, as knowledge of tumor behavior progressed, Hanahan and Weinberg proposed additional emerging hallmarks involved in deregulation of cell metabolism, escape from immune destruction, induction of tumor inflammation, genome instability and mutations. Currently, new evidence supports rapidly evolving fields that can be considered as additional emerging hallmarks of cancer, such as the contribution of stem cell programs that can promote cancer initiation, therapy resistance and metastasis [35,36].

Hereafter, we describe a curated list of exceptionally well-characterized lncRNAs, published in journals with impact factors around ten and higher. We analyzed and annotated their described mechanisms of action and the main hallmarks of cancer they enable to promote HCC progression. We believe this helps put lncRNA research in context with current efforts to pharmacologically target the acquired malignant capabilities of liver cancer cells and we suggest that the development of lncRNA-based therapies as stand-alone or combinatorial agents could potentially improve the current suboptimal outcomes of HCC patients.

### 2.1. Activating Invasion and Metastasis

LncRNA deregulation leading to the activation of invasion and metastasis is the most represented hallmark of cancer in our main lncRNA collection (Figure 2). This is not surprising, as HCC shows a high tendency to metastasize and infiltrate adjacent and distant tissues [37]. In fact, the overall poor prognosis of HCC is largely the result of its metastatic nature and the high rate of tumor recurrence after surgery, which remain the main causes of lethal outcomes in HCC [38].

Epithelial-to-mesenchymal transition (EMT) is an essential event in the early stages of invasion required for cell escape and colonization, and therefore is highly regulated in normal cells by several pathways, including the transforming growth factor-β (TGF-β) pathway [106]. TGF-β exerts antiproliferative effects on normal and premalignant cells. However, advanced cancers often become insensitive to the tumor-suppressive actions of TGF-β and instead, benefit from TGF-β’s profound metastasis-promoting effects, such as EMT induction [107]. The EMT prometastatic response to TGF-β is mediated by several transcription factors (ZEB1/2 (zinc finger E-box binding homeobox 1/2), SNAIL1/2 (Snail family transcriptional repressor 1/2), TWIST, SLUG and members of the FOX (forkhead box) family) and other downstream effectors, including cytokines (interleukins (IL)-6, IL-11), growth factors (VEGF), matrix proteins (fibronectin, collagen), cytoskeletal (vimentin) and cell adhesion proteins (E- and N-cadherin), metalloproteases (MMPs) and more recently described factors such as microRNAs (miRNAs) [108] and lncRNAs [109]. LncRNAs are able to promote and sustain molecular events leading to metastasis by enabling several possible mechanisms, predominantly by binding miRNAs (Figure 2) and working as competing endogenous RNAs (ceRNAs) or by regulating more intricate lncRNA–miRNA protein axes.

**LncRNA-ATB** (activated by TGF-β) is a paradigmatic ceRNA, which increases invasion and metastasis by sequestering members of the miR-200 family (miR-200a, b and c), known EMT regulators that target ZEB1 and ZEB2 [79,110]. Accordingly, in vivo models of LncRNA-ATB upregulation showed increased intrahepatic dissemination, higher levels of circulating tumors cells (CTCs) and multiple-site metastasis. In addition, other mechanisms besides miR binding work to promote metastasis in lncRNA-ATB-expressing cells. LncRNA-ATB binds to and increases the stability of IL-11 mRNA, improving IL-11 secretion and activating STAT3 signaling in an autocrine manner (Figure 1). This confers a survival advantage to metastatic cells [111].

Similarly, many other lncRNAs regulate downstream transcription factors related to canonical TGF-β-mediated EMT. In some cases, the mechanism also involves miRNA binding, such as **ZFAS1** (zinc finger antisense 1) [105] which upregulates ZEB1 by sequestering miR-150. LncRNA **MUF** (mesenchymal stem cells upregulated factor) [89] also acts as a ceRNA for miR-34a, resulting in increased SNAIL transcription and EMT activation. This is in accordance with previous results in colon, breast and lung carcinomas where decreased miR-34 expression was found to de-repress SNAIL transcription [112]. Additional lncRNAs upregulate the expression of EMT transcription factors through unknown mechanisms. LncRNA **MITA1** (metabolism-induced tumor activator 1) [88], induced by energy stress, increases SLUG transcription. LncRNA **HCCL5** (hepatocellular carcinoma L5) [51] which is transcriptionally regulated by ZEB1 via a super-enhancer, is a general positive regulator of EMT genes SNAIL, SLUG, TWIST and ZEB1 in a positive feedback loop. Of these, SNAIL and TWIST have been directly reported to induce EMT, invasion and metastasis in HCC [113,114].

Cells that undergo EMT reorganize their cytoskeleton to enable cell elongation and directional motility [107]. LncRNA **DREH** (downregulated expression by hepatitis B virus X protein) [45], represses EMT, likely by binding directly to vimentin to change normal cytoskeleton structure. Other lncRNAs affect Rho GTPases, which control the process of actin remodeling in EMT [115]. By competitively binding to miR19a, lncRNA **HOXD-AS1** (homeobox D cluster antisense 1) [58] de-represses Rho GTPase activating protein 11a (ARHGAP11A), while **lncMER52A** (lncRNA derived from MER52A retrotransposons) [78] stabilizes the cell–cell adhesion regulator p120-catenin (Figure 1). Both proteins are involved in the activation of the Rho GTPases, RAC1 (Ras-related C3 botulinum toxin substrate 1) and CDC42 (cell division cycle 42), which promote the formation of membrane protrusions and cell motility, and the initiation of front-rear polarity. All of these are essential events in EMT that enable directional migration. Finally, an interesting lncRNA is **PXN-AS1** (paxillin (PXN) antisense RNA 1). PXN is a cytoskeletal protein located in sites of cell adhesion to the extracellular matrix and has been involved in cancer cell proliferation and invasion. PXN mRNA stability and translation are regulated by PXN-AS1 in a dual manner. When the oncofetal splicing factor MBNL3 is expressed, it induces PXN-AS1 exon 4 inclusion, resulting in increased PXN and tumor promotion. This occurs as PXN-AS1 containing exon 4 binds to the 3´UTR region of PXN mRNA and impedes the negative action of miR-24. Instead, PXN-AS1 lacking exon 4 binds to coding sequences of PXN mRNA and decreases translation. Therefore, MBNL3 knockdown almost completely abolishes HCC growth [80].

As cells acquire motility and invasive capacities, additional factors such as metalloproteinases (MMPs) help the cell move through the surrounding stroma towards a blood or lymphatic vessel potentiating invasiveness [116]. This process can also be enabled by lncRNA–miRNA protein axes. For example, **HOXD-AS1**, this time by binding to miR-130a, prevents miRNA-dependent degradation of the SOX4 mRNA transcription factor, activating the expression of target genes EZH2 and MMP2 [59]. Similarly, **ZFAS1**, besides ZEB1, is able to induce MMP14 and MMP16, two membrane MMPs with relevant roles in HCC metastasis and invasion [117,118,119].

Several other studies have found lncRNAs to promote or protect against metastasis in HCC cells by less frequent mechanisms involving regulatory signaling pathways such as Wnt-β-catenin (lncRNA **MUF**) [89], NF-κB (lncRNA **miR503HG** (MIR503 host gene)) [87] or novel metastasis-related factors such as FBXW7 (lncRNA **CASC2** (circulating cancer susceptibility 2)) [40] with varying degrees of evidence of clinical relevance. While MUF, MITA1 or **PTENP1** (phosphatase and tensin homolog pseudogene 1) [96] lack validation in HCC tissues (Figure 3), other lncRNAs such as miR503HG, CASC2, ZFAS or lncRNA-ATB have validated correlations with overall survival (OS) and importantly, with micro and macrovascular invasion or portal vein tumor thrombus (PVTT), one of the first events in intrahepatic dissemination (Figure 4). These clinical correlations help validate a potential role for these lncRNAs in the invasion and metastasis of HCC and support their targetability with therapeutic purposes.

### 2.2. Sustaining Proliferative Signaling

The proliferation and survival of normal hepatocytes is under the control of several layers of regulation owing to its remarkable ability to regenerate. However, alterations in signaling pathways arising from mutations or epigenetic or epitranscriptomic changes can render cells independent or refractory to the control of these pathways. In most HCC patients, such alterations are the result of sustained damage exerted by HCC risk factors. Exposure to these factors leads to persistent liver inflammation and progressive liver fibrosis that often leads to cirrhosis. Liver cirrhosis generates a pro-tumorigenic microenvironment that promotes the transformation of pre-malignant lesions into highly proliferating neoplastic foci, a mechanism that accounts for up to 90% of HCC cases [120]. This process is initiated by activating mutations in the telomerase (TERT) gene, considered a gatekeeper of early hepatocarcinogenesis. TERT expression allows cirrhotic hepatocytes to override their senescent state and reactivate proliferation [121]. Furthermore, transcriptomic analyses of premalignant and early HCC samples show that MYC activation also plays a central role in early malignant transformation [122]. This process is strongly enhanced by mutations occurring in one or more oncogenes or tumor suppressors, predominantly β-catenin activation and p53 inactivation. Additional, less frequent mutations add up to a cumulative mutation burden of up to 70 mutations within each tumor [123,124]. This outstanding mutational burden impacts several pathways including PI3K/Akt, Ras/Raf/MEK/ERK, TGF-β, Wnt or NOTCH. All play important roles in supporting HCC growth, frequently promoting invasion and metastasis concomitantly as co-occurring malignant traits.

LncRNAs seem to follow this trend as there is a significant overlap between lncRNAs involved in invasion/metastasis and proliferation. The combination of these two hallmarks accounts for the majority of lncRNAs in our main collection (Figure 2) and even more in our larger collection (Appendix A). Despite this overlap, the mechanisms of action induced by lncRNAs that affect the proliferative potential of HCC cells seem to differ from those of lncRNAs involved in promoting invasion/metastasis. While miRNA binding is also prominent, lncRNAs involved in the malignant proliferation of HCC cells act by varied mechanisms, including binding to effector proteins and epigenetic and novel epitranscriptomic regulators.

LncRNA **SNHG10** (small nucleolar RNA host gene 10) [100] is a good example of an lncRNA working at the intersection between proliferation and metastasis induction. By sponging miRNA-150 and stabilizing the mRNA of RPL4 (receptor-like protein 4), SNHG10 acts to increase the expression and activity of the MYB (myeloblastosis) proto-oncogene. Reciprocally, MYB enhances SNHG10 expression by promoter activation in a positive feedback loop. Further, SNHG10 modulates the expression of its homolog SCARNA13 (small Cajal body-specific RNA 13) through the miR-150/RPL4/MYB pathway. Then, SCARNA13 upregulates the levels of the transcription factor SOX9 (SRY-box transcription factor 9) with repercussions in HCC cell growth and metastasis. This regulatory reinforcement may be behind the drastic phenotype of SNHG10 depletion in subcutaneous and orthotopic mouse models as well as the outstanding correlation of SNHG10 expression with patient prognosis. Indeed, SNHG10 is the lncRNA that correlates with more clinical parameters in our lncRNA collection (Figure 4).

**LINC00662** (long intergenic non-protein coding RNA 662) [68] activity is more specific to the induction of HCC proliferation by mediating genome-wide hypomethylation through depletion of SAM (S-adenosylmethionine) and increase in SAH (S-adenosylhomocysteine) levels. The ratio of the methionine cycle intermediates SAM and SAH is commonly regarded as an indicator of the methylation potential of the cell, and changes in its balance have been implicated in liver carcinogenesis [125,126]. LINC00662 binds to the 3′-UTR of MAT1A (S-adenosylmethionine synthase isoform type-1) mRNA to promote its decay and reduce the levels of MAT1A, a liver-specific SAM synthesizing enzyme. At the same time, LINC00662 interacts with and promotes the ubiquitination and degradation of the enzyme AHCY (adenosyl homocysteinase), which hydrolyzes SAH and prevents its accumulation. Therefore, by RNA–RNA and RNA–protein interactions, LINC00662 is able to decrease the SAM/SAH balance and de-represses methylated oncogene promoters, activating known HCC oncogenes such as MYC, CTNNB1 and HRAS.

By enabling similar effector-binding mechanisms, lncRNAs **UFC1** (ubiquitin-fold modifier conjugating enzyme 1) [104] and **uc.134** (long non-coding RNA uc.134) **[103]**, up and downregulated in HCC, respectively, are also involved in HCC proliferation through oncogene activation. In a multistep regulatory network, lncRNA uc.134 binds to CUL4 (cullin-4A), inhibiting its nuclear export and preventing CUL4-mediated ubiquitinylation and degradation of LATS1 (large tumor suppressor kinase 1), a tumor suppressor. This increases the stability and overall levels of LATS1 which in turn phosphorylates and inactivates the YAP1 (yes1-associated transcriptional regulator) oncoprotein, effectively silencing its target genes, among them c-MYC. Conversely, in a more straightforward mechanism, lncRNA UFC1 is able to bind with the mRNA stabilizing protein HuR (Hu antigen R) to induce the levels of β-catenin mRNA, favoring proliferation of HCC cells and the growth of subcutaneous mouse tumors. An unexpected mechanism of action for an lncRNA is that of **LINC00998** (long intergenic non-protein coding RNA 998) [71], that contains an open reading frame and is translated to a small peptide, SMIM30, which promotes HCC growth and metastasis. SMIM30 is required for membrane anchoring and functionality of tyrosine kinases SRC/YES1 leading to MAPK pathway activation (Figure 1). In turn, **HNF1A-AS1** (hepatocyte nuclear factor 1 homeobox A antisense RNA 1) is a target of HNF1A, considered a strong tumor suppressor for HCC [53]. HNF1A-AS1 mediates the protective role of HNF1A by binding and increasing the activity of the phosphatase SHP1 (protein tyrosine phosphatase non-receptor type 6), a strong inhibitor of JAK/STAT, NF-κB and AKT pathways [53,127].

Other mechanisms requiring lncRNAs for HCC cell proliferation include epigenetic and epitranscriptomic changes. Epigenetic regulation in HCC has been gaining momentum over the past few years as a promising target that may be actionable using epigenetic drugs [128]. Instead, epitranscriptomic changes have only been recently described to be relevant for HCC progression [129]. **ANRIL** (cyclin-dependent kinase inhibitor 2B (CDKN2B) antisense RNA (1) [39], a known regulator of the polycomb repressive complex 2 (PRC2), was found to bind and recruit PRC2 to the promoter of the differentiation factor KLF2 (Krüppel-like factor (2) in HCC cells, effectively repressing KLF2 transcription to promote HCC cell growth in vitro and in vivo. **GATA3-AS** (GATA-binding protein 3) [46] has been shown to regulate m6A deposition on the pre-mRNA of the tumor suppressor GATA3. To do this, GATA3-AS base pairs with GATA3 pre-mRNA and guides KIA1429 (a scaffolding component of the m6A methyltransferase complex) binding. This increases m6A marks in the 3′UTR of GATA3 pre-mRNA that prevent binding of HuR (Figure 1), promote RNA degradation, reduce GATA3 levels and help HCC proliferation and metastasis.

While there is a clear role of lncRNAs in regulating epigenetic or epitranscriptomic events and there is preclinical evidence of their potential from mouse xenografts, many of these lncRNAs, including ANRIL or GATA3-AS, fail to validate their relevance in patient survival or as prognostic factors (Figure 3 and Figure 4). Further efforts need to be dedicated to evaluating their therapeutic potential for HCC patients.

### 2.3. Deregulating Cell Metabolism

In order to enable malignant cell proliferation, adjustments in cell metabolism have to be made that increase the biomass and provide the energy required for cell division and growth. Interestingly, instead of using oxidative phosphorylation as the main energy source for dividing, tumor cells utilize glycolysis even when enough oxygen is available [130]. This phenomenon is known as the Warburg effect and it usually involves genetic mutations that increase glucose intake. Some lncRNAs have been involved in this process (Figure 2). Similar to ANRIL, **TUG1** (taurine upregulated gene 1) epigenetically represses KLF2 transcription by binding PRC2 and recruiting it to the KLF2 promoter [101]. In addition, TUG1 sequesters miR-455-3p, which targets AMPKβ2 mRNA (AMP-activated protein kinase containing the β2 regulatory subunit), leading to an increase in HK2 (hexokinase 2), a rate-limiting factor required for the first step in most glucose metabolism pathways [102]

Interestingly, many lncRNAs have been described to regulate the PI3K/AKT (phosphatidylinositol-3-kinase (PI3K)/protein kinase B(AKT)) pathway in HCC. AKT induces mTOR activity, which promotes glucose metabolism, lipid and nucleotide biosynthesis and protein translation. **CASC9** (circulating cancer susceptibility 9) [131] forms a functional cytoplasmic complex with HNRNPL (heterogeneous nuclear ribonucleoprotein L) that regulates AKT function [131]. Likewise, **PTTG3P** (pituitary tumor-transforming 3, pseudogene) [97] is upregulated in HCC and activates PI3K/AKT signaling [97]. **MALAT-1** (metastasis-associated lung adenocarcinoma transcript 1) also leads to activation of the mTOR pathway by modulating the splicing targets of the oncogenic splicing factor SRSF1 (serine- and arginine-rich splicing factor 1), which also promotes the production of anti-apoptotic splicing isoforms [84,85]. **HULC** (HCC upregulated lncRNA) activates the AKT–PI3K–mTOR pathway after PTEN (phosphatase and tensin homolog) inhibition through miR-15a/P62. In addition, HULC also promotes methylation of the miR-9 promoter and suppresses the targeting of PPARA (peroxisome proliferator-activated receptor alpha), which activates the ACSL1 (Acyl-CoA synthetase 1) promoter and induces lipogenesis. HULC also contributes to HCC growth by phosphorylation of YB-1 (Y-box-binding protein 1), which leads to the release of YB-1 from its bound mRNA and translation of silenced oncogenic mRNAs [60,61,62,63]. Finally, by sequestering miR-327, HULC allows increased levels of PRKACB (protein kinase cAMP-activated catalytic subunit beta), which phosphorylates CREB (cAMP response element-binding) transcription factor, leading to higher levels of CREB target genes such as cFOS or HULC itself [64]. An additional inducer of nucleotide metabolism is **LincNMR** (long intergenic non-coding RNA-nucleotide metabolism regulator) [72], that binds to YBX1 and locates to the promoter of RRM2 (ribonucleotide reductase subunit M2), TYMS (thymidylate synthetase) and TK1 (thymidine kinase 1) genes, increasing the expression of these essential enzymes for nucleotide synthesis [72].

Other modulators of lipid metabolism are **Linc00958** (long intergenic non-coding RNA 00958) and **NEAT1** (nuclear paraspeckle assembly transcript 1). Linc00958 functions by sponging miR-3619-5p, increasing HDGF (hepatoma-derived growth factor) expression and resulting in lipogenesis and progression of HCC [70]. Instead, NEAT1 binds to miR-124-3p and allows expression of the adipose triglyceride lipase (ATGL), highly expressed in human HCC. High ATGL, increases the levels of diacylglycerol and free fatty acids and PPARA (peroxisome proliferator-activated receptor alpha) signaling, which works as a predictor of poor prognosis [91].

### 2.4. Self-Renewal and Maintenance of Cancer Stem Cells

Liver cancer stem cells (LCSCs) can derive from genetic or epigenetic alterations of liver progenitor cells or hepatocytes involved in regeneration or dedifferentiation processes [132]. Thus, stem or progenitor cells can also be the cells of origin of certain HCCs [35]. In fact, many studies suggest that embryonic liver and HCC development share similar alterations in their genetic programs. More importantly, HCC patients with gene expression profiles similar to stem cells show a worse prognosis [133,134]. LCSCs impact cell division and reprogramming but they also affect tumor heterogeneity, relapse, metastasis, and drug resistance. Induction, self-renewal, proliferation, and differentiation of LCSCs are complex processes regulated by a myriad of signaling pathways including JAK/STAT, NF-κB, Wnt/β-catenin, Notch, Hedgehog, Hippo/TAZ/YAP or JNK.

LncRNAs have been shown to regulate HCC by influencing LCSCs. For example, although with certain controversy, lncRNA **H19** has been involved in the deregulation of LCSCs in different tumors, including HCC where it contributes to doxorubicin resistance [135,136]. Many LCSC lncRNA regulators affect the activity of chromatin remodeling complexes SWI–SNF (SWItch/sucrose non-fermentable), INO80 (INO80 complex ATPase subunit) or NuRD (nucleosome remodeling deacetylase) complex. **LncTCF7** (transcription factor 7)**/WSPAR** (Wnt signaling pathway activating non-coding RNA) binds SWI–SNF to promote expression of TCF7, a major transcription factor in the Wnt signaling pathway in LCSCs [82]. Of interest, TCF7 is barely detected in normal liver, which makes it a promising biomarker in HCC [82]. In the case of **LncBRM** (long non-coding RNA for association with Brahma), the interaction is with the SWI–SNF subunit BRM (Brahma), resulting in the activation of YAP1 (yes1-associated transcriptional regulator) signaling [75]. Similarly, **lncRNA HAND2-AS1** (heart and neural crest derivatives expressed 2- antisense RNA 1) recruits the INO80 chromatin-remodeling complex to the promoter of BMPR1A (BMP (bone morphogenetic protein) receptor type 1A), thereby inducing its expression and leading to the activation of BMP signaling which activates the initiation of HCC [49]. **LncHDAC2** (lncRNA histone deacetylase 2), highly expressed in LCSCs, recruits the NuRD complex to the promoter of PTCH1 (protein-patched homolog 1), impairing PTCH1 expression. This induces the hedgehog signaling pathway and promotes the self-renewal of LCSCs [77].

Instead, other lncRNAs affect the function of major regulators including NF-κB (nuclear factor kappa-beta), STAT3 (signal transducer and activator of transcription 3) or β-catenin. **DILC** (downregulated in liver cancer stem cells) functions as a tumor suppressor to inhibit LCSCs by preventing NF-κB binding, inhibiting IL-6 transcription and abolishing JAK2/STAT3 pathway activation. Downregulation of DILC in HCC allows stimulation of LCSCs [43,137,138]. **Lnc-Sox4** (SRY-box transcription factor 4), upregulated in HCC cells, interacts with the transcription factor STAT3 and recruits it to the promoter of Sox4 to stimulate its transcription. Sox4 overexpression then leads to initiation of LCSCs [81]. Interestingly, **Lnc-β-Catm** (lncRNA for β-catenin methylation) recruits the PRC2 subunit EZH2 (enhancer of zeste homolog 2) and contributes to EZH2-mediated methylation of β-catenin. This impedes β-catenin ubiquitination and degradation and supports the self-renewal of LCSCs through the Wnt pathway [74]. Similarly, **LINC00210** (long intergenic non-protein coding RNA 210), highly expressed in LCSCs, binds to CTNNBIP1 and blocks its inhibitory activity over β-catenin, resulting in activation of the Wnt pathway and promoting self-renewal [67]. β-catenin expression is also regulated at the mRNA level by **DANCR** (differentiation antagonizing non-protein coding ribonucleic acid). DANCR binding to the 3´UTR of CTNNB1 mRNA impedes the repressing effect of miR-214, 320a and 199a (Figure 1), increasing β-catenin expression and the stemness features of HCC cells [42]. Additionally, acting at the mRNA level, **ICR** (intercellular adhesion molecule 1 (ICAM-1) related) increases the stability of ICAM-1 mRNA through RNA duplex formation [65]. This is relevant as ICAM-1 is an adhesion molecule highly expressed in LCSCs that plays pivotal roles in carcinogenesis and metastasis.

### 2.5. Resisting Cell Death

Programmed cell death by apoptosis is considered as an inherent boundary to halt cancer development. Tumoral cells try to attenuate the signaling circuits operating in the apoptotic program to successfully override death cues [35]. A major master regulator of apoptosis is the tumor suppressor p53, whose gene shows inactivating mutations in 12–48% of HCCs [123]. In line with the high relevance of this factor, there are several lncRNAs that function to increase or decrease p53 activity in several tumors including HCC. **Lnc-lp53** (lncRNA induced by p53) sits among the negative regulators [73]. Lnc-lp53 attenuates p300-acetylating activity on p53 and binds to HDAC1 (histone deacetylase 1), preventing its degradation and maintaining deacetylated/inactive p53, altogether abrogating p53 activity. Instead, **PRAL** (P53 regulation-association long non-coding ribonucleic acid), downregulated in HCC, facilitates the interaction of HSP90 and p53, inhibiting MDM2-dependent p53 ubiquitination and resulting in enhanced p53 stability [93]. Similarly, **PSTAR** (P53-stabilizing and activating RNA) binds to hnRNPK (heterogeneous nuclear ribonucleoprotein K) and enhances its SUMOylation, thereby strengthening the interaction between hnRNPK and p53, which ultimately leads to the accumulation and transactivation of p53 [94].

Other well-known apoptosis regulators are factors required for development, as they need to coordinate several processes, including differentiation, motility, and apoptosis. This includes HOXA genes, homeodomain transcription factors whose expression is induced by **HOTTIP** (*HOXA* transcript at the distal tip). HOTTIP is a pro-tumoral *cis*-acting lncRNA upregulated in HCC that activates transcription of the *HOXA* locus by local recruitment of the positive regulator WDR5–MLL (WD repeat-containing protein 5–mixed linage leukemia) complex [29,57]. **MCM3AP-AS1** (minichromosome maintenance deficient 3-associated protein antisense RNA 1) is involved in HCC cell proliferation by targeting miR-194-5p and subsequently promoting FOXA1 (forkhead box A1) expression in HCC cells [86]. FOXA1 is a pioneer factor and a well-known positive regulator of liver specific gene expression with a fundamental role in impairing liver cell apoptosis.

Other factors may affect apoptosis, including NF-κB. While the role of NF-κB in cancer by regulating immunity and inflammation is well documented, there are some reports that link NF-κB with tumor progression by regulating apoptosis [139], a function of NF-κB that may be mediated in part by lncRNA PDIA3P1 (protein disulfide isomerase family A member 3 pseudogene 1). PDIA3P1 is involved in a regulatory axis with NF-κB, by preventing miR-124 and miR-125 targeting of TRAF6. This leads to activation of the pathway and downstream anti-apoptosis genes, promoting doxycycline-induced apoptosis in vitro and sensitizing mouse xenografts to doxycycline treatment [92].

### 2.6. Enabling Replicative Immortality

The unlimited proliferative potential of tumor cells as a way to achieve immortality constitutes one of the main keys of cancer. The ability to maintain telomeres protecting the ends of chromosomal DNAs to avert apoptosis is achieved by regulation of telomere expression by different mechanisms [35]. This couples to a remodeling of the cell cycle machinery to promote cell division and restrain checkpoint controls. Several lncRNAs function as regulators of cell-cycle factors. **HEIH** (high expression in HCC) and **DLEU2** (deleted in lymphocytic leukemia 2) modulate EZH2 activity. LncRNA-HEIH enhances EZH2 for post-transcriptional regulation of cyclin-dependent kinase inhibitor genes (p15, p16, p21 and p57), playing a key role in G0/G1 phase of the cell cycle [52]. DLEU2 is upregulated in HCC and linked to HBV. HBx and DLEU2 co-recruitment on the cDNA displaces EZH2 from the viral chromatin to boost transcription and viral replication. In addition, DLEU2-HBx association with target host promoters relieves EZH2 repression and leads to transcriptional activation of the cyclin B2 gene, among others [44].

Other lncRNAs that affect the cell cycle at different levels are **LALR1** (liver regeneration 1), which recruits CTCF (CCCTC-binding factor) to repress AXIN1 (protein phosphatase 1) promoter, facilitating cyclin D1 expression through Wnt/β-catenin activation [66]. **LncUCID** (lncRNA upregulating CDK6 by interacting with DHX9), enhances CDK6 (cyclin-dependent kinase 6) expression by competitively binding to DHX9 (ATP-dependent RNA helicase A) and sequestering DHX9 from CDK6-3’UTR [83]. Finally, **PVT1** (plasmacytoma variant translocation 1) upregulates cell cycle genes by increasing the stability of nucleolar protein NOP2 [98].

### 2.7. Inducing Angiogenesis

The tumor-associated neovasculature, due to the permanent activation of angiogenesis, constitutes another main key for HCC progression [35]. However, few lncRNAs from our main collection have been associated with angiogenesis. One example is **lncRNA- MVIH** (microvascular invasion in HCC), which is upregulated in HCC, and favors the formation of new vessels and, therefore, HCC metastasis. MVIH represses the secretion of PGK1 (phosphoglycerate kinase 1), an inhibitory regulator of angiogenesis, although the detailed mechanism is still unknown [90]. Another lncRNA related to angiogenesis is **H19** (imprinted maternally expressed untranslated transcript) which was the first lncRNA described back in 1990 and it is downregulated in HCC [140]. The reason for this is that H19 is activated by Sox2 (transcription factor SOX-2), and Sox2 is inhibited by TGF-β, whose levels are high in the tumor microenvironment. Therefore, there is a significant negative correlation between the TGF-β gene signature and H19 function [47]. In spite of this, LCSC-derived exosomes contain high levels of H19 which is thought to induce angiogenesis in endothelial cells [48]. In fact, an indirect manner to promote angiogenesis is through exosome secretion. **HOTAIR** (HOX transcript antisense RNA) accelerates MVB (multivesicular bodies) transport by inducing Ras-related protein Rab-35 (RAB35) expression and localization to MVBs. Further, HOTAIR promotes MVB fusion by regulating the colocalization of vesicle-associated membrane protein 3 (VAMP3) and synaptosome-associated protein 23 (SNAP23) [54,55]. This results in increased exosome secretion, which may contribute to tumor growth by several means and provides a useful source of biomarkers for tumor diagnosis and prognosis.

Finally, lncRNA regulation can also impact on the levels of hypoxia-inducible factors (HIF), whose function is sensing and reacting against hypoxia by triggering angiogenesis. At normal oxygen levels, the alpha subunit of the HIF transcriptional complex (HIF1A) is targeted to degradation by prolyl hydroxylation carried out by EGLN2 (prolyl hydroxylase 1) [141]. Interestingly, a 4-bp insertion/deletion polymorphism (rs10680577), located in a distal promoter of EGLN2, affects **RERT**-lncRNA (RAB4B-EGLN2 read-through long non-coding RNA) structure and expression, which correlates positively with EGLN2 levels [99]. The deletion allele was significantly associated with increased risk of HCC and this association was more significant in current smokers.

### 2.8. Tumor-Promoting Inflammation and Avoiding Immune Destruction

Despite their strong relevance for HCC, very few lncRNAs have been described with a role in tumor-promoting inflammation or in avoiding immune destruction. This is the case of **LINC00665** and **Lnc-EGFR** (long non-coding epidermal growth factor receptor). LINC00665 binds to the double-stranded RNA (dsRNA)-activated protein kinase (PKR), blocks its degradation by ubiquitination and increases PKR activity [69]. This activates NF-κB signaling, which contributes to LINC00665 expression, resulting in a positive feedback loop. Interestingly, LINC00665 contributes to enhanced cell proliferation in vivo and to poor prognosis of HCC patients. Instead, Lnc-EGFR is expressed in T regulatory cells (Tregs), correlating positively with tumor size and EGFR expression and negatively with IFN-γ levels. Lnc-EGFR binds to EGFR and blocks its degradation by ubiquitination by the ubiquitin ligase c-CBL (Casitas B-lineage lymphoma). Increased EGFR levels activate AP1/NF-AT1 (activator protein 1/ nuclear factor of activated T cell 1) transcription, resulting in a positive feedback loop that increases EGFR levels and in the promotion of Treg differentiation that contributes to the immunosuppressive state of HCC [76].

## 3. Concluding Remarks

This review is the result of a strong effort that has involved extensive examination of hundreds of lncRNAs related to HCC (Appendix A) and the selection of few representative studies (Table 1) with the aim of classifying them according to their mechanism of action (Figure 2), associated cancer hallmarks (Figure 2 and Figure 3) and correlation with relevant clinical parameters (Figure 3 and Figure 4) to find potentially useful associations with therapies that are currently in the clinic or that are expected to arrive in the near future (Figure 5). The classification into cancer hallmarks should be taken with flexibility, as several biological processes involved in tumor progression play relevant roles in overlapping hallmarks and lncRNAs are not an exception. In fact, similar to protein multifunctionality, there are several lncRNAs that follow different mechanisms to impact on single or multiple hallmarks. Examples of this are HOXD-AS1 and ZFAS1 function in invasion and metastasis, HULC in cell metabolism and proliferative signaling or H19 in both the maintenance of cancer stem cells and angiogenesis. Clinical parameters chosen were OS and medical or histological features linked to worse prognosis, such as tumor stage, vascular invasion, tumor size, degree of histological differentiation, presence of metastasis or AFP levels (Figure 4). Significant associations between these clinical parameters and the expression of HCC-related lncRNAs give them a non-negligible degree of clinical relevance. Moreover, some lncRNAs have been studied in both massive public databases and in independent cohorts of HCC patients, which confers these candidates a higher level of evidence for clinical translation (Figure 3).

### 3.1. Molecular Mechanisms of lncRNAs Relevant for HCC

Despite the identification and publication of large numbers of lncRNAs in the literature, our understanding of their functions is still limited [142]. However, we believe that the collection discussed in this review is a good representation of a larger collection of HCC-relevant lncRNAs and helps clarify the major mechanisms enabled by lncRNAs in HCC cells (Appendix A). LncRNAs can interact with proteins, DNA or other RNAs to modulate gene expression or protein activity. By far, most lncRNAs are described to function primarily as ceRNAs or miRNAs sponges (Figure 2 and Appendix A). This over-representation of such a mechanism of action has generated controversy in the field for several reasons. On one hand, miRNAs are generally expressed to higher levels than lncRNAs. However, ceRNA function should require equimolar or higher levels of lncRNA versus the target miRNA, as the great majority of lncRNAs described to function as ceRNAs (including those described in this review) harbor but one binding site for their suggested miRNA target. The experiments performed for the identification of ceRNAs do not normally take this issue into consideration, as they generally involve ectopic overexpression of the lncRNA and miRNA components. The functional analysis of miRNA-binding site deletion mutants, as has been done in the case of LncRNA-ATB, together with a careful quantification of the molecules present in the cell, should be performed to ensure ceRNA functionality. On the other hand, each mRNA can be targeted by multiple miRNAs and each miRNA is predicted to regulate hundreds of targets. Therefore, the sequestration of a single miRNA by a lncRNA should impact gene expression at several levels and this may vary in different cells depending on their miRNAome. Similar concerns can be applied to lncRNAs described to bind and control the activity of proteins regulators. Examples include factors that regulate mRNA splicing or stability (SRSF1, HuR, hnRNPs) or protein phosphorylation, methylation, acetylation, or ubiquitination (SHP1, EZH2, p300, HDAC1, CUL4). Most studies fail to address whether lncRNA activity over these general regulators can reach certain specificity and how this is achieved. Otherwise, lncRNA expression should affect all targets of these global regulators, and lead to the deregulation of several cellular pathways.

A clearer picture is possible for lncRNAs that bind directly to a specific factor, rather than to a global regulator. LncRNA binding to a specific mRNA may impede miRNA blockade or HuR regulation or may modulate mRNA translation and stability by other molecular mechanisms. Similarly, lncRNA binding to a specific genomic locus may control chromatin remodeling or epigenetic modifications, or the activity of transcription factors impacting on the expression of a specific gene. The ability of lncRNAs to bind mRNAs involves interactions by base pairing and therefore it may be easily acquired evolutionarily just by gaining the expression of genes located antisense to the target gene. Likewise, naïve induction of the expression of lncRNAs that may function in *cis* co-transcriptionally is an excellent way to control local gene expression at the genomic level. All these are exclusive lncRNA features that should contribute to the relevance of lncRNAs in processes where evolutionary pressure plays a fundamental role. This is the case of carcinogenesis, cancer cell establishment and proliferation in the tumor microenvironment, tumor evasion from the immune system and tumor escape from anticancer therapies. In accordance, therapies targeting such lncRNAs may show unprecedented beneficial effects.

### 3.2. Therapeutic Implications

The most promising attempt at lncRNA inhibition therapy at the RNA level is grounded on antisense oligonucleotides (ASOs), which can efficiently reduce lncRNA levels [143]. Transcriptional deregulation or depletion by genome editing can also be achieved with CRIPSR-Cas technologies. Currently, the use of ASOs seems the more feasible strategy to drug oncogenic lncRNAs, as their effects are reversible in nature and their development is simplified due to their uniform chemical process. Although the pharmacodynamic and pharmacokinetic properties of ASOs are unconventional and remain to be fully described, ASO-based therapies, such as Nusinersen (SPINRAZA^®^) for treatment of spinal muscular atrophy, have already been safely used in patients [144]. Alternatively, restoration of tumor suppressor lncRNAs that are downregulated in HCC has several additional barriers to overcome before becoming a potentially useful strategy for HCC treatment. However, the current COVID-19 pandemic has shown, at world scale, the efficacy and safety of lipid nanoparticles and viral vectors for the delivery of RNA or DNA. Lipid nanoparticles, in particular, appear to be a natural and safe way to deliver lncRNAs or ASOs into cells with potential high organ-specific targetability [145]. Once the last frontiers involving delivery and safety issues are broadly surpassed, it is expected that lncRNA targeting will be widely used as a therapy to treat several diseases, including cancer. This could be done as single or combination therapies.

Regarding combination therapies, nowadays, the main HCC molecular targets with commercialized drugs are immune targets (PD-1, PD-L1, CTLA-4), VEGF or multi-TK (tyrosine kinases) inhibitors (Figure 5). Anti-VEGF therapies or anti-TK exert their therapeutic effect by targeting multiple hallmarks of cancer such as angiogenesis, proliferation, evasion of growth suppressors and cell death or activating invasion and metastasis. As described before, some lncRNAs, such as TUG1, MVIH, PVT1 or HOTTIP, have been related to these hallmarks with a significant impact on patient survival. Therefore, combined therapies with both anti-lncRNA and a direct protein target could boost the tumor targeting, increase response rates and impact patient management. Interestingly, other possible coding targets from the enabling replicative immortality, the self-renewal and maintenance of CSCs or the deregulating cell energetics hallmarks do not have any available drugs. These include highly relevant un-druggable factors for HCC such as the tumor suppressor p53 or the oncogenic β-catenin. Since many lncRNAs have been described that modulate the activity of these factors or with a related hallmark function and with effect over OS, many potential targets arise.

Remarkably, only lnc-EGFR from our selection and an additional study on lncRNA FENDRR [146] have been found to regulate immune escape in HCC. However, evidence from studies in other tumors have found clinically relevant lncRNAs associated with antigen release [147], antigen presentation [148,149,150,151], immune cell activation [152] and importantly, with induction of PD-L1 expression and immune escape [153,154]. Therefore, efforts to study lncRNAs involved in immune checkpoint regulation in HCC could potentially help to find new therapeutic targets and potentiate the outstanding improvement in response rates obtained with drugs affecting such signaling pathways. Similarly, other missing lncRNAs in fields of therapeutic relevance are those involved in regulating genome stability or TERT function. Interestingly, it has been recently found that lncRNA NIHCOLE (non-coding RNA involved in HCC with an oncogenic role in ligation efficiency), an lncRNA highly expressed in HCC that associates with OS and several clinical parameters, functions to increase the ligation efficiency of DNA double-strand breaks through the NHEJ pathway of DNA repair (Unfried et al., under review). This provides a new avenue for therapeutic intervention. On the same line, inhibition of TERT with ASOs has been recently shown to impact on HCC growth in model animals. However, clinical translation of this result needs to ensure local delivery of the therapy to guarantee that TERT is not inhibited in tissues other than HCC [155].

Future research in HCC-related lncRNAs is also expected to identify novel candidates that function through the expression of micropeptides. Although contradictory, it is now widely accepted that some RNAs annotated as non-coding harbor small open reading frames (ORFs). The explanation is simple. Short ORFs may occur randomly in the genome and may represent regions that are not truly translated, and therefore they are not considered as actual ORFs in genomic annotations. Nevertheless, microprotein translation is noticeable in the liver and may also extensively occur in liver cancer [156]. Interestingly, some microproteins are secreted or are expressed as transmembrane proteins, as is the case of SMIM30 [71]. Therefore, their activity could be neutralized with specific antibodies. In the case of micropeptides that function as tumor suppressors, their functionality could be mimicked with synthetic peptides that may be introduced in the cell fused to cell-penetrating sequences. In addition, the small domain that constitutes the micropeptide is amenable for the selection of small molecules or specific aptamers that impair functionality, which may have therapeutic implications for oncogenic micropeptides.

### 3.3. Biomarker Potential

Current efforts to analyze molecular and genetic information from tumor samples are essential for identifying new drug targets, but also new biomarkers to assess patient prognosis, disease progression or treatment selection and follow-up. Nonetheless, this requires a tissue sample which provides only static information of the tumor features at the time of extraction. Further, procedures to biopsy tumors are not always feasible in HCC patients, and conventional clinical HCC biomarkers (AFP, AFP lectin fraction or des- γ- carboxy prothrombin) lack sensitivity and specificity. This underscores the need to investigate more effective markers of disease stage and progression. Given the high HCC specificity of some lncRNAs, they appear as promising biomarker candidates. This has been extensively reviewed recently [157,158]. In fact, the studies that have analyzed lncRNA biomarkers in HCC show that they can be detected in extracellular vesicles (EVs), circulating immune cells or as cell-free lncRNAs in the plasma or serum of HCC patients. This would allow the use of liquid biopsies to enable periodic monitoring of disease status with minimally invasive specimens such as serum or plasma [159]. However, despite cumulating evidence of the biomarker potential of circulating HCC-lncRNAs, further research efforts are required. The clinical implementation of circulating lncRNA HCC biomarkers, either by quantifying multiple-lncRNA panels alone or in combination with other biomarkers such as miRNAs and/or AFP, may drastically impact the assessment of HCC disease in the near future.

In summary, lncRNAs play essential roles in different hallmarks of HCC, becoming excellent putative therapeutic targets for precision medicine. It should be clarified that several studies support that only a fraction of all lncRNAs are functionally relevant [160,161,162,163]. However, in the case of HCC, some of those lncRNAs seem to confer a fitness advantage to cancer cells, offering a unique opportunity to expand the repertoire of potential targets. This is essential for HCC that activates several undruggable pathways [121] and where novel therapies have shown potent responses but only in a fraction of patients [164]. HCC patients could benefit from the additive or synergistic effects of co-targeting with novel first-line treatments and potentially useful lncRNA-based drugs. Finally, in addition to targets for precision medicine, some lncRNAs could be excellent biomarkers for diagnosis and patient stratification. Screening tissue or liquid biopsies for the presence and abundance of HCC-lncRNAs or prognostic lncRNA signatures, together with the therapeutic targeting of lncRNAs, could soon become standard clinical practices, providing a much-needed tool to meet the challenges of HCC treatment.

## Figures and Tables

**Figure 1 cancers-13-02651-f001:**
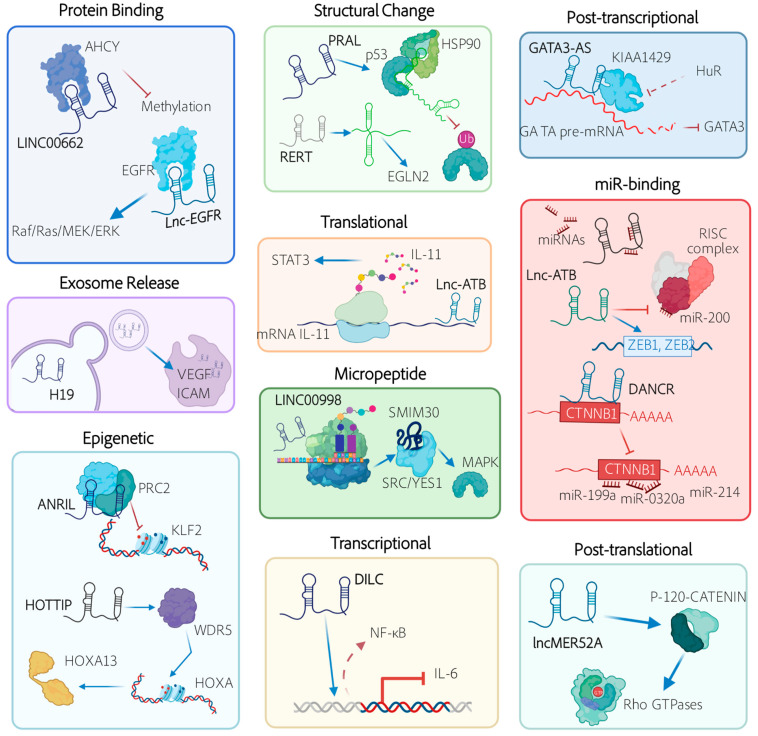
Mechanisms of action enabled by lncRNAs in HCC. LncRNAs can work at several levels to impact HCC development and progression. See text for details. LncRNA-linked blue arrows depict activation while red arrows indicate inhibition.

**Figure 2 cancers-13-02651-f002:**
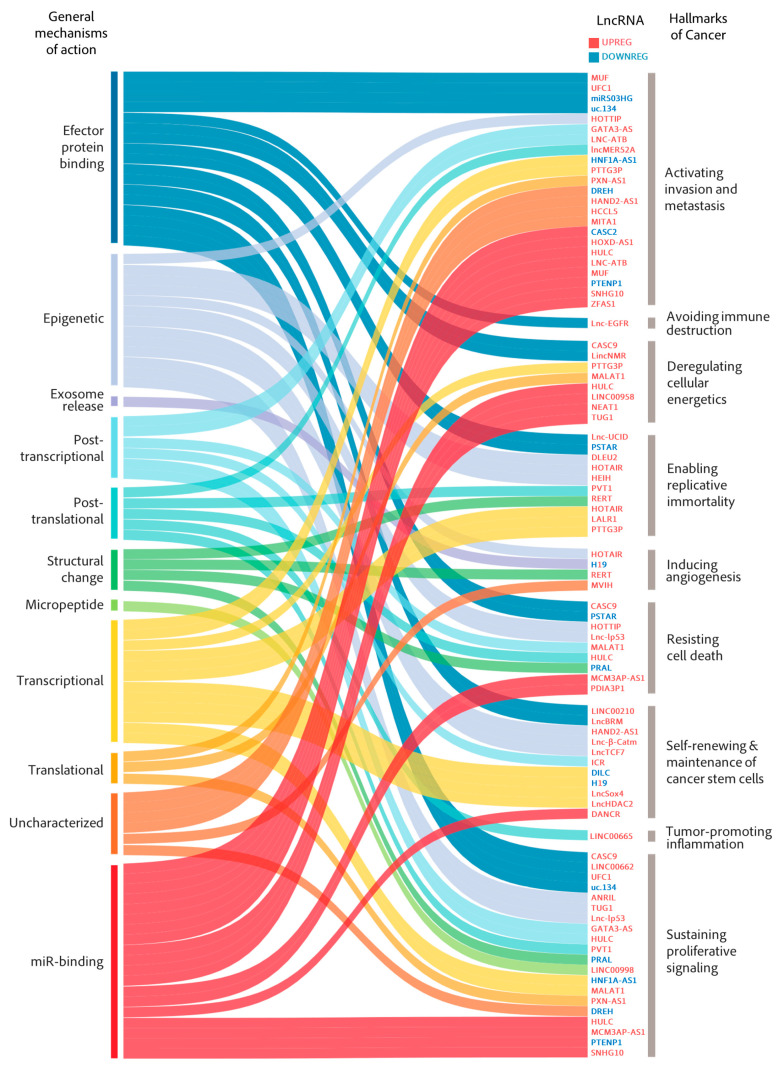
Sankey diagram linking mechanisms of action and hallmarks of cancer of HCC-relevant lncRNAs. LncRNAs selected according to scientific significance, preclinical validation and/or association with relevant clinical parameters have been classified according to their mechanism of action (left) and the cancer hallmark they modulate (right). Note that some lncRNAs have been assigned to several mechanisms of action and/or different cancer hallmarks. LncRNAs downregulated in HCC are shown in blue while those upregulated are in red. The full list of candidates in shown in Table 1.

**Figure 3 cancers-13-02651-f003:**
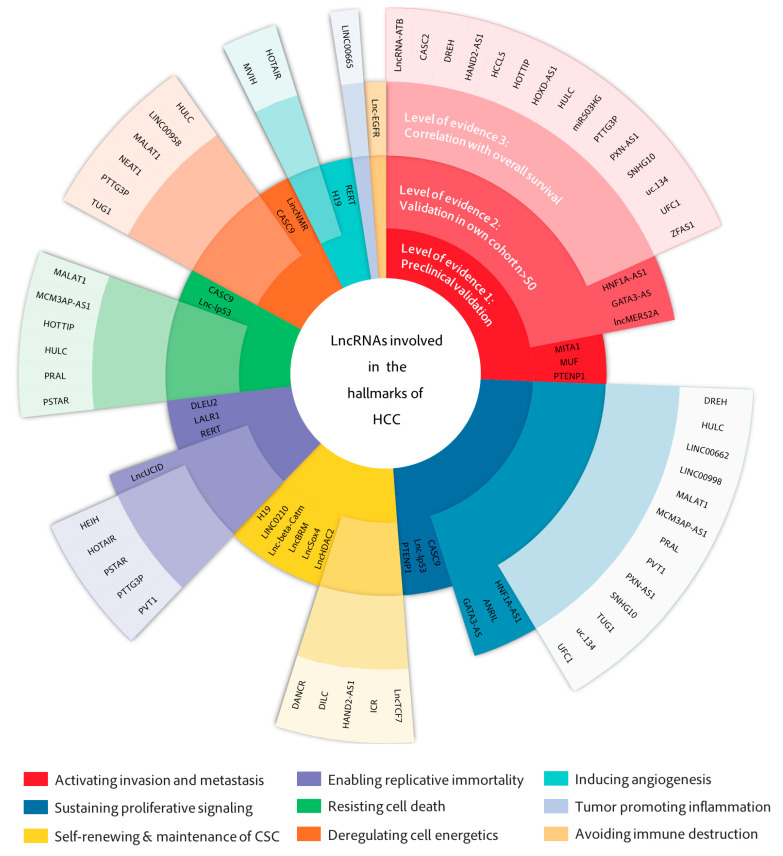
Degree of in vivo validation of HCC lncRNAs stratified according to hallmarks of cancer. LncRNAs described in Figure 1 and classified according to the hallmarks of cancer they modulate have been stratified in three steps called “levels of evidence” of their clinical relevance. Level 1 requires preclinical validation in model animals and has been done for all cases. Level 2 indicates that lncRNA expression and/or clinical associations have been studied in at least two independent cohorts of patients (*n* > 50). Level 3 is granted when the expression of the lncRNA has a significant association with overall survival.

**Figure 4 cancers-13-02651-f004:**
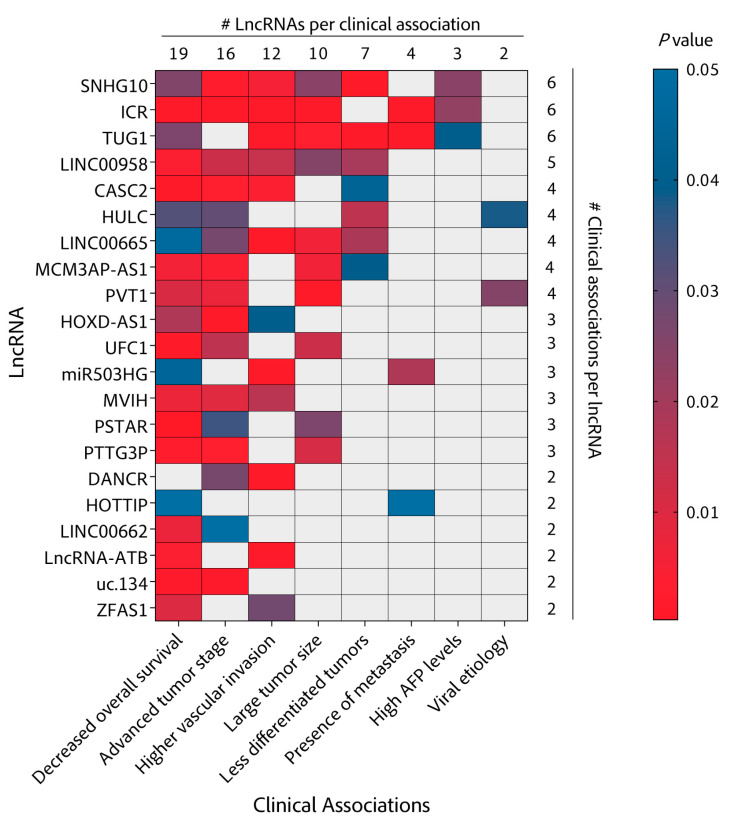
Heatmap of clinical associations. LncRNAs (left) with at least two clinical associations have been ordered according to the number of significant associations from top (highest) to bottom (lowest). Clinical associations have been also ordered from left to right. The numbers of lncRNAs per clinical association (top) and the clinical associations per lncRNA (right) are also indicated. Significances (*p*-values) were extracted from the publications describing each lncRNA and are indicated with a color gradient described to the right of the heatmap.

**Figure 5 cancers-13-02651-f005:**
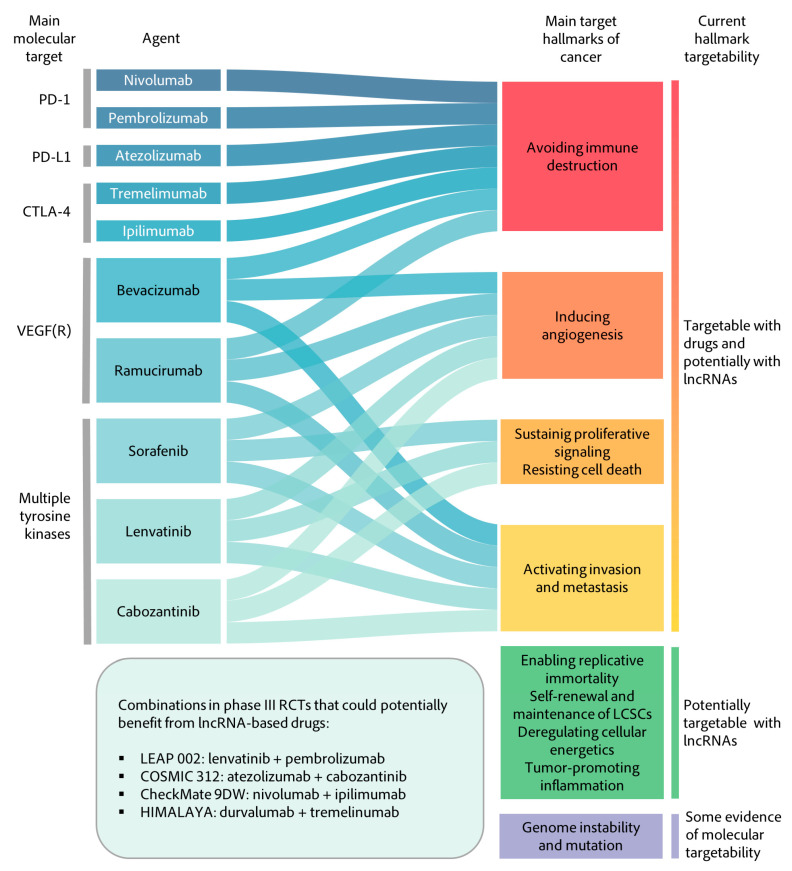
Sankey diagram of suggested combination therapies. Approved drugs for HCC and their main molecular targets are shown to the left and linked to their main cancer hallmark and the hallmark druggability, including those based on potential lncRNA targeting. The box at the bottom shows ongoing Phase III clinical trials that could also benefit from lncRNA therapy.

**Table 1 cancers-13-02651-t001:** List of lncRNAs described in this review. “Biochemistry” shows the molecular mechanism of the lncRNA (target and detail). “Pre-Clinical” describes the cell lines and the animal model used. “Clinical” shows the number of patients studied in an independent cohort and the significance of the association between the expression of the lncRNA and the overall survival. LncRNA names in red denote upregulated lncRNAs, whereas blue denotes downregulated lncRNAs. All candidates are in alphabetical order.

LncRNA	Biochemistry		Pre-Clinical		Clinical	
	Target	Detail	Cell Type	Model	Own Cohort (*n* > 50)	OS (*p* Value)
ANRIL [39]	↑PRC2	Epigenetically represses KLF2 by binding PRC2 and recruiting it to KLF2 promoter	HepG2	SC	77	
CASC2 [40]	↓miR-367	Sponges miR-367 and prevents targeting of FBXW7	MHCC-97H, Hep-3B	M	75	*p* < 0.001
CASC9 [41]	↑HNRNPL	Regulates AKT function and DNA damage sensing	HLE	CAM	No HCC tissues	
DANCR [42]	↑CTNNB1	Associates with CTNNB1 and blocks the repressing effect of miR-214, 0320a and 199a on CTNNB1	SMMC7721	SC, OT	135	*p* = 0.0003
DILC [43]	↓IL-6	Binds to IL-6 promoter, preventing NF-κB binding and inhibits IL-6 transcription, IL-6/STAT3 signaling and LCSC expansion	LM3	SC	195	*p* = 0.019
DLEU2 [44]	↑DLEU2-HBx	DLEU2-HBx association with target host promoters relieves EZH2 and leads to transcriptional activation	No animal model		No HCC tissues	
DREH [45]	↓Vimentin	Binds and represses vimentin to change normal cytoskeleton structure and inhibit tumor metastasis	Hepa1-6	SC, OT, M	100	*p* = 0.039
GATA3-AS [46]	↓GATA3	Interacts with KIAA1429 for binding to GATA3 pre-mRNA, competitively preventing binding of HuR and promoting the degradation of GATA3 pre-mRNA	SK-Hep1, HCCLM3	SC, M	70	Not with lncRNA but KIAA1429
H19 [47,48]	↑VEGF, ICAM	CSC-like CD90+-derived exosomes contain high levels of H19 which induces angiogenesis in endothelial cells	No animal model		No exosomes	
	↑TGF-β	TGF-β inhibits expression of Sox2 TF and Sox2-mediated activation of H19 in TICs	TICs	OT	No HCC tissues	
HAND2-AS1 [49,50]	↑INO80	Recruits the INO80 chromatin-remodeling complex to the promoter of BMPR1A, inducing its expression and activation of BMP signaling	HUH7, patient-derived samples	SC, DEN	60	*p* < 0.05 from TCGA
		Screening of recurrently deregulated lncRNAs enriched in co-expressed clusters of genes related to cell adhesion	No animal model		No HCC tissues	
HCCL5 [51]		Transcriptionally driven by ZEB1 via a super-enhancer. Activated by TGF-β. Upregulates EMT genes	SMMC-7721	SC	196	*p* < 0.05 from TCGA
HEIH [52]	↓EZH2	Binds to EZH2, inhibiting EZH2 targets such as p15, p16, p21 and p57	HUH7, SMMC-7721	SC	50/107/85	*p* = 0.014
HNF1A-AS1 [53]	↑SHP-1	Interacts with SHP-1, increasing phosphatase activity	HUH7, MHCC-LM3	SC, M	277	Not with lncRNA but HNF1
HOTAIR [54,55,56]	↑RAB35, VAMP3-SNAP23 and p-SNAP23	Promotes MVB fusion by regulating the location and phosphorylation of SNAP23 to form the SNARE complex	No animal model		No exosomes	
	↑SUZ12	Together with DDX5, regulates SUZ12 stability and PRC2-mediated gene repression of EpCAM, pluripotency genes and HBV cccDNA-encoded genes.	No tumor: hepatocyte-specific Hnf4a-null mice		52	
	↑miR218	Activates P16Ink4a and P14ARF signaling, increasing miR-218 expression and suppressing Bmi-1, resulting in inhibition of P14 and P16	HepG2	SC	No HCC tissues	
HOTTIP [57]	↑WDR5	Upregulates oncogenic transcription factors such as HOXA13	HepG2	SC	52	
HOXD-AS1 [58,59]	↓miR-130a-3p	Binds miR-130a-3p and prevents SOX4 miRNA-mediated degradation, thus activating the expression of EZH2 and MMP2	HUH7	M	120	*p* = 0.0179
	↓miR19a	Upregulates Rho GTPase ARHGAP11A by competitively binding to microRNA-19a	HCCLM3	SC, M	60	Not with lncRNA but ARHGAP11A
HULC [60,61,62,63,64]	↑p-YB1	Promotes the phosphorylation of YB-1 and release from silenced oncogene mRNAs	No animal model		41	*p* = 0.032
	↓IGF2BP1	Acts as an adaptor protein of the CCR4-NOT that destabilizes HULC	No animal model		60	
	↓miR-15a	Inhibits PTEN through miR-15a/P62, activating AKT-PI3K-mTOR	Hep3B	SC	30	
	↓miR-9	Promotes methylation of miR-9 promoter and suppresses targeting of PPARA, which activates the ACSL1 promoter and lipogenesis	HepG2, HUH7	SC	60	
	↓miR-372	Inhibits miR-372, derepresses PRKACB, which induces CREB phosphorylation and HULC transcription in an auto regulatory loop	No animal model		14	
ICR [65]	↑ICAM-1	Increases the stability of ICAM-1 mRNA through RNA duplex formation	HUH7	SC	245/372	*p* < 0.001
LALR1 [66]	↑CTCF	Recruits CTCF to repress AXIN1 promoter, increasing cyclin D1 expression through Wnt/β-catenin activation	Liver transfection		No HCC tissues	
LINC00210 [67]	CTNNBIP1	Interacts with CTNNBIP1 and blocks its inhibitory role in Wnt/β-catenin activation.	Liver TICs	SC	5	
LINC00662 [68]	↑AHCY ↓MAT1A	Regulates MAT1A and AHCY that influence SAM and SAH levels to maintain genomic hypomethylation	HCCLM9	SC, M	70	*p* = 0.0071
LINC00665 [69]	↑PKR	Interacts with PKR, enhances its activation and blocks degradation, resulting in a positive regulation of NF-κB	HUH7	SC	50/122	*p* = 0.0472
LINC00958 [70]	↓miR-3619-5p	Sponges miR-3619-5p to upregulate HDGF expression, facilitating lipogenesis and progression	Primary HCC cells	SC, OT	80	*p* = 0.0039
LINC00998 [71]	↑SRC/YES1	Codes for the SMIM30 micropeptide that binds SRC/YES1, to drive anchoring and phosphorylation, activating MAPK pathway	HUH7	SC	369/160	*p* = 0.00014
LINCNMR [72]	↑YBX1	Binds YBX1 which regulates RRM2, TYMS and TK1 expression binding to their promoter regions	HLE	CAM	No HCC tissues	
Lnc lp53 [73]	↑HDAC1, p300	Interacts with HDAC1 and p300 to prevent HDAC1 degradation and attenuate p300 activity, resulting in increased CDKN1A and PUMA	SK-N-SH	SC	No HCC tissues	
Lnc-beta-Catm [74]	↑EZH2	Associates with β-catenin and EZH2, promoting β-catenin, methylation and reduced ubiquitination, thus promoting its stability, leading to activation of Wnt–β-catenin	Primary HCC cells	SC	No HCC tissues	
LncBRM [75]	↓miR-150	Binds miR-150 and abrogates its tumor-suppressive function by inhibiting ZEB1, MMP14 and MMP16	Primary HCC cells	SC	6	Not with lncRNA but YAP1
Lnc-EGFR [76]	↑EGFR	Binds to EGFR inhibiting its interaction with c-CBL and blocking its ubiquitination and sustaining its activity	DCs, CD4+, CD8+ T cells and 97H	SC	125	
LNCHDAC2 [77]	NuRD or ↓PTCH1	In liver CSCs, recruits the NuRD onto PTCH1 promoter to inhibit expression, and activation of Hedgehog signaling		SC	3/6	Not with lncRNA but HDAC
lncMER52A [78]	↑ p120-catenin	Stabilizes p120-catenin and triggers the activation of Rho GTPase and p120-ctn/Rac1/Cdc42 axis	HUH7	M	120	*p* > 0.05
LncRNA-ATB [79]	↓miR-200	Sponges miR-200s family derepressing ZEB1 and ZEB2	SMMC-7721, HCCLM6	OT	86	*p* = 0.004
	↑IL-11	Binds and stabilizes IL-11 mRNA, causing autocrine IL-11 induction and STAT3 signaling-mediated colonization				
lncRNA-PXN-AS1 [80]	↑PXN mRNA	MBNL3 induces lncRNA-PXN-AS1 exon 4 inclusion, which allows PXN-AS1 to bind to PXN mRNA preventing its degradation	SMMC-7721, QSG-7701, HUH7	SC	279	0.0402
LncSox4 [81]	↑STAT3	Interacts with and recruits Stat3 to the Sox4 promoter to initiate the expression of Sox4	Primary HCC cells	SC	No HCC tissues	
lncTCF7 [82]	↑SWI/SNF	Recruits SWI/SNF complex and together they activate TCF7 expression. TCF7 activates Wnt signaling, priming LCSC self-renewal	HEP3B, HUH7	SC	37	*p* < 0.05
LncUCID [83]	↓DHX9	Enhances CDK6 expression by competitively binding to DHX9 and sequestering DHX9 from CDK6-3’UTR	QGY-7703	SC	139	
MALAT1 [84,85]	↑Wnt	Through SRSF1, regulates alternative splicing of effector genes (↑RPS6KB1, TEAD1, mTORC1, ↓ BIM, BIN1 ↑Cyclin D1)	PHM-1	SC	35	
		Prevents gluconeogenesis and promotes glycolysis	No animal model		No HCC tissues	
MCM3AP-AS1 [86]	↓miR-194-5p	Targets miR-194-5p and subsequently promotes FOXA1 expression	Hep3B	SC	80	*p* = 0.0054
miR503HG [87]	↓ HNRNPA2B1	Interacts with HNRNPA2B1 and promotes its degradation, decreasing the stability of p52 and p65 mRNA and suppressing the NF-κB signaling	HUH7, SMMC-7721	OT	93	*p* = 0.045
MITA1 [88]		Promotes EMT, partially by increasing Slug transcription	SK-Hep1	M	No HCC tissues	
MUF [89]	↓miR-34a	Works as a ceRNA for miR-34a, leading to Snail1 upregulation and EMT	SMMC-7721	SC,M	No HCC tissues	
	↑ANXA2	Binds Annexin A2 (ANXA2) and activates Wnt/β-catenin signaling				
MVIH [90]	↓PGK1	Prevents PGK1 secretion	HCCLM3	SC, OT, M	40/215/65	*p* = 0.007
NEAT1 [91]	↑miR-124-3p	Binds miR-124-3p to regulate ATGL expression	HUH7, MHCC-LM3	SC, OT	29/40	*p* = 0.0236
PDIA3P1 [92]	↓miR-125a/b/miR-124	Binds to miR-125a/b/miR-124 and relieves their repression over TRAF6, leading to activation of NF-κB pathway	QGY-7703	SC	347	
PRAL [93]	↑HSP90, p53	Stem-loop motifs at the 5’end of lncRNA-PRAL facilitate union of HSP90 and p53, inhibiting MDM2-dependent p53 ubiquitination, resulting in enhanced p53 stability	SMMC-7721	SC	56/189/102/80	*p* = 0.035 (c.2) *p* = 0.027 (c.3)
PSTAR [94]	↑hnRNPK	Binds to hnRNPK and enhances its SUMOylation and interaction between hnRNPK and p53, resulting in accumulation and transactivation of p53	HepG2	SC, OT	38/127/136	*p* = 0.0003
PTENP1 [95,96]	↓miR-17, 19b and 20a”	Decoys oncomiRs miR-17, miR-19b and miR-20a, preventing targeting of PTEN, PHLPP and autophagy genes ULK1, ATG7 and p62	Mahlavu	SC	No HCC tissues	
	↑miR-21	Exosomes increase miR-21 and downregulate PTEN, PTENP1 and TETs, promoting tumor growth	SNU-449	SC	No HCC tissues	
PTTG3P [97]	↑PTTG1	Upregulates PTTG1, activates PI3K/AKT signaling and its downstream signals	HepG2	SC	90	*p* = 0.002
PVT1 [98]	↑NOP2	Increases stability of nucleolar protein NOP2	SMMC-7721	SC	89	*p* = 0.0104
RERT [99]	↑EGLN2	rs10680577 affects RERT-lncRNA structure and subsequently EGLN2 expression	No animal model			
SNHG10 [100]	↓miR-150-5p	Sponges miR-150-5p and interacts with RPL4 mRNA to increase the expression and activity of c-Myb. Enhanced SNHG10, SCARNA13 and SOX9 expression.	SNU-387 and HCCLM3	SC, OT	64	*p* = 0.0255
TUG1 [101,102]	↑PRC2	Epigenetically represses KLF2 transcription by binding PRC2 and recruiting it to KLF2 promoter region	HepG2	SC	77	
	↑miR-455-3p	Induces miR-455-3p, which targets the 3’UTR of AMPKβ2, which downregulates HK2 promoting glycolysis and metastasis	SK-Hep1	M	239/242	*p* = 0.026 (c.1); *p* = 0.039 (c.2)
uc.134 [103]	↑CUL4	Binds to CUL4A, inhibits its nuclear export and ubiquitination of LATS1. Increases p-YAP to silence its target genes	MHCC97-H and HCCLM3	SC, M	170	*p* < 0.001
UFC1 [104]	↑HuR	Interacts with HuR to increase levels of β-catenin mRNA and protein	SK-Hep1, BEL-7402	SC	131	*p* < 0.001
ZFAS1 [105]	↓miR-150	Binds miR-150 and abrogates its tumor-suppressive function over ZEB1, MMP14 and MMP16	HUH7	SC, M	113	*p* < 0.01

(SC) Subcutaneous, (M) Metastasis, (OT) Orthotopic, (CAM) Chorioallantoic Membrane, (DEN) diethylnitrosamine, (c.) cohort, (miR) microRNA, (TCGA) The Cancer Genome Atlas.

## Data Availability

All data needed to evaluate the conclusions in the paper are present in the paper and/or the Appendix A.

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
