# Peer review of "The Landscape of lncRNAs in Hepatocellular Carcinoma: A Translational Perspective"

_cancers, 2021, doi:10.3390/cancers13112651_

Round 1

Reviewer 1 Report

In this review, Unfried and Sangro et al. describe the implications of lncRNAs for the hallmarks of cancer in the context of hepatocellular carcinoma. The review is well written, the topic is relevant in the field and the effort conducted by the authors to summarize the literature should be praised.

However, there are few suggestions for the authors:

1) It should be added a topic with a briefly description of long non-coding RNAs’ synthesis, structure and mechanisms of function.

2) Since about 80% of HCC are related to HBV or HCV infection, a short paragraph discussing interactions between lncRNAs and viral oncoproteins that could be important for the hallmarks of cancer could be added.

3) A short material and methods section could be included explaining how the papers for the review were selected, how figures were constructed and p values for figure 3 were obtained.

4) At concluding remarks, it also should be included a chapter discussing their implications as biomarkers.

Author Response

REVIEWER ONE

In this review, Unfried and Sangro et al. describe the implications of lncRNAs for the hallmarks of cancer in the context of hepatocellular carcinoma. The review is well written, the topic is relevant in the field and the effort conducted by the authors to summarize the literature should be praised.

We would like to thank the reviewer for his/her excellent work, as we believe he/she has helped to improve the quality of our manuscript.

However, there are few suggestions for the authors:

1) It should be added a topic with a briefly description of long non-coding RNAs’ synthesis, structure and mechanisms of function.

The reviewer is right that this description would clarify the review for those readers that are not lncRNA experts. We have now included a brief description of lncRNA synthesis, structure and mechanisms of action. See the revised version of the manuscript (tracked changes) in page 2 (second paragraph) and 3 (first and second paragraphs).

2) Since about 80% of HCC are related to HBV or HCV infection, a short paragraph discussing interactions between lncRNAs and viral oncoproteins that could be important for the hallmarks of cancer could be added.

We thank the reviewer for this suggestion. The world of lncRNAs in HCV and HBV infection is complex. We now mention that infection with HCV and HBV leads to deregulation of several lncRNAs that affect viral oncoproteins and may increase hepatocarcinogenesis. We briefly describe the example of EGOT and we cite two excellent reviews about HBV and HCV lncRNAs and their links to liver diseases. This can be found in page 3 (third paragraph).

3) A short material and methods section could be included explaining how the papers for the review were selected, how figures were constructed and p values for figure 3 were obtained.

We thank the reviewer for this suggestion as this will help underlie our claim of the relevance of the studies selected. However, since this is not a systematic review, a methods section per se is not contemplated in the review format of Cancers. Nonetheless, we have made changes throughout the manuscript that we believe clarify all the reviewer concerns raised in this point.

  • We now describe that we selected studies published in journals with an impact factor around ten and higher at the end of the introduction. This can be found in section two last paragraph.
  • On the construction of the figures, we have changed the name of former figures 1 and 4 (figures 2 and 5 in the revised manuscript) to better explain the type of graph used to represent the data i.e. Sankey diagrams, in both cases. See figure legends for figures 2 and 5. Additionally, in the Acknowledgements section, we have detailed the software employed for the design of each figure.
  • To clarify the origin of p values for figure 3 (figure 4 in the revised version) we have now added a clarification sentence where we state that p values were extracted from each publication. See the figure legend for figure 4 in the revised manuscript.

4) At concluding remarks, it also should be included a chapter discussing their implications as biomarkers.

The reviewer is right to recommend an extended description of lncRNA biomarker potential, as this is relevant for the translational perspective of the manuscript. We have now added an additional section (section 3.3 in the revised manuscript) in the concluding remarks that briefly discusses the potential of lncRNA biomarkers in HCC and their advantages over current options, and we suggest two excellent recent reviews that expand on this matter as a reference for the readers.

Reviewer 2 Report

This is a very good review, which I strongly recommend for publication.

However a bit of due diligence could make it better:

1) Grammar in Table 1: either use present or past indefinite in ALL cases (present looks better). There are quite frequent misspellings: e.g. LINC00665 interacts with PKR... and BLOCK degradation (blocks its degradation); lncMER52A stabilizes (not atabilizes) p12-catenin; PDIA3P1 - Binds (not binds), etc.

2) Other misspellings in the text: Section 2.7 "HOTAIR promotes MVB fusion and...And what?; Next to last sentence in section 3.1 - "tumor scape" - escape? (the same misspelling in section 3.2). Finally some sentences are too long and should be cut into two or even three sentences.

3) But to the heart of the matter: figures are good, but in one area they are insufficient. In my opinion, verbal explanations of molecular interactions strain readers' imagination. My non-obligatory advice consists in making a figure, which in several panels depicts all molecules interacting with several chosen lncRNAs. This will help to comprehend the essence and will increase citation. If you agree - I will send my comments in a day or two, if not - that's OK as it is.

Author Response

REVIEWER TWO

This is a very good review, which I strongly recommend for publication.

We would like to thank the reviewer for his/her work and his/her kind words

However a bit of due diligence could make it better:

1) Grammar in Table 1: either use present or past indefinite in ALL cases (present looks better). There are quite frequent misspellings: e.g. LINC00665 interacts with PKR... and BLOCK degradation (blocks its degradation); lncMER52A stabilizes (not atabilizes) p12-catenin; PDIA3P1 - Binds (not binds), etc.

We thank the reviewer for bringing our attention to this issue. We have now extensively proofread Table 1 for the misspellings, and format and grammar errors pointed out by the reviewer and additional errors found upon careful inspection of Table 1 text.

2) Other misspellings in the text: Section 2.7 "HOTAIR promotes MVB fusion and...And what?; Next to last sentence in section 3.1 - "tumor scape" - escape? (the same misspelling in section 3.2). Finally some sentences are too long and should be cut into two or even three sentences.

We thank the reviewer for their careful examination of the manuscript, which will undoubtedly increase the readability of this review. We have corrected all the errors pointed out by the reviewer and we have looked throughout the manuscript for additional errors and for difficult-to-read sentences. We have now made several small changes in misspelling, grammar and paragraph construction that we believe improve the use of language and the overall readability of the manuscript.

3) But to the heart of the matter: figures are good, but in one area they are insufficient. In my opinion, verbal explanations of molecular interactions strain readers' imagination. My non-obligatory advice consists in making a figure, which in several panels depicts all molecules interacting with several chosen lncRNAs. This will help to comprehend the essence and will increase citation. If you agree - I will send my comments in a day or two, if not - that's OK as it is.

We agree with the reviewer that the many biochemical pathways described in the review can be hard to follow and we are glad to add the suggested figure to the revised version of the manuscript. To reduce the number of lncRNAs to describe we selected those with the best clinical correlations, and we now feature their regulatory pathways divided by the mechanism of action they enable to exert such regulation. See figure 1.